# Prevalence and Risk Factors of Menstrual Disorders in Korean Women

**DOI:** 10.3390/healthcare13060606

**Published:** 2025-03-11

**Authors:** Ye-Lin Kim, Jun Young Chang, Suejin Kim, Mira Yoon, Jae-Na Ha, Kang Hyun Um, Boeun Lee, Kyoung Sook Jeong

**Affiliations:** 1Department of Environmental Health Research, National Institute of Environmental Research, Incheon 22689, Republic of Korea; yelin@korea.kr (Y.-L.K.); acmecjy@korea.kr (J.Y.C.); suenier@korea.kr (S.K.); sswu77@korea.kr (M.Y.); freya0127@korea.kr (J.-N.H.); 2Yonsei Geopyoung Medical Checkup Center, Seoul 06121, Republic of Korea; um12345@naver.com; 3Department of Occupational and Environmental Medicine, Wonju Severance Christian Hospital, Institute of Occupational & Environmental Medicine, Wonju College of Medicine, Yonsei University, Wonju 26426, Republic of Korea

**Keywords:** menstruation, menstrual cycle, menstrual disorders, polymenorrhea, oligomenorrhea, menorrhagia, reproductive health, women’s health

## Abstract

Background: Some women experience menstrual disorders such as polymenorrhea, oligomenorrhea, and menorrhagia, which are not only influenced by biological factors but also by lifestyle and psychosocial factors. Understanding menstrual disorders is essential for women’s health and quality of life. Objectives: To identify policies that are needed to prevent menstrual disorders, we investigated the prevalence and risk factors of menstrual disorders in this study. Methods: A web-based questionnaire survey evaluated menstrual characteristics and biological, lifestyle, and psychosocial risk factors in 13,943 South Korean females aged 15–45 years. A Chi-square test was used to compare the prevalence of menstrual disorders by general and psychosocial characteristics. A logistic regression analysis was utilized to determine odds ratios (ORs) of risk factors for menstrual disorders. Adjusted ORs of the risk factors for menstrual disorders, after adjusting for other risk factors, were calculated. Results: The prevalence of polymenorrhea, oligomenorrhea, and menorrhagia was 3.1%, 9.0%, and 5.4%, respectively. A significantly high prevalence of menstrual disorders was associated with the following risk factors: underweight (OR: 1.291) and current smoking (OR: 1.516) for polymenorrhea; overweight (OR: 1.354), obesity (OR: 2.164), current drinking (OR: 1.170), depression (OR: 1.416), and perceived stress (OR: 1.248) for oligomenorrhea; and depression (OR: 1.521) for menorrhagia. Conclusions: This cross-sectional study highlighted that menstrual disorders are significantly associated with lifestyle habits and psychosocial factors in South Korean women. These findings can serve as scientific evidence to support public health initiatives aimed at enhancing awareness and menstrual health management among women.

## 1. Introduction

Menstrual health is closely related to women’s reproductive health, influencing fertility rates and child health. Healthy pregnancies and childbirth increase neonatal survival rates and enable healthier childcare provision, contributing to population structure stability [1]. Reportedly, menstrual health is related to various diseases, such as endocrine (metabolic disease) and cardiovascular diseases [2,3], highlighting its association with women’s health status and quality of life. Irregular or prolonged menstrual cycles are associated with an increased risk of premature mortality due to cardiovascular disease and cancer [4] and may contribute to the development of non-alcoholic fatty liver disease [5]. Thus, menstrual health has a significant impact on women’s overall health. Improvements in women’s health can accelerate economic growth by increasing productivity and expanding educational opportunities [6]. Therefore, women’s health should not only be considered on an individual level but should be managed from a public health perspective for societal development and stability [6].

Menstruation is a natural physiological phenomenon that occurs once a month in women of childbearing age, indicating healthy reproductive system performance. Menstrual cycle length typically ranges from 21 to 35 days, and the menstrual duration lasts between 2 and 7 days [7]. However, some women experience irregular menstrual cycles, including polymenorrhea (menstrual cycle length shorter than 21 days), oligomenorrhea (menstrual cycle length longer than 35 days), and menorrhagia (menstruation lasting longer than a week) [8]. A study utilizing data from the National Health Insurance Service–National Health Screening Cohort from 2009 to 2016 indicated that the prevalence of menstrual disorders among South Korean women (15–49 years) increased from 8.6% in 2009 to 11.6% in 2016 [9]. Furthermore, according to a recent health survey on sexual and reproductive health in South Korea, 26.5% of 2137 women aged 19–64 years reported experiencing menstrual disorders, such as irregular menstruation and excessive bleeding [10]. In a study involving 538 Korean adolescents aged 14–18 years, 32.3% of respondents reported having irregular menstrual cycles [11]. Additionally, another study utilizing the Korea National Health and Nutrition Examination Survey from 2010 to 2012 reported that 15% of 801 adolescents had experienced irregular menstrual cycles [12]. Despite these findings, South Korean women tend to avoid or delay visits to obstetrics and gynecology clinics. This may be influenced by the conservative nature of the Confucianism-based society in Korea that often associates such clinic visits—particularly for young, unmarried women—with negative social perceptions [13]. Consequently, reproductive health conditions such as menstrual disorders may remain undetected in their early stages, increasing the risk of related diseases. Therefore, raising awareness about the importance of addressing menstrual disorders is essential.

Menstrual disorders can occur due to menstrual cycle complications, influenced not only by biological factors, such as hormonal imbalances, but also by lifestyle (body mass index [BMI], smoking status, and alcohol consumption status) and psychosocial (stress and depression) factors. Many studies have corroborated findings regarding the impact of BMI, smoking, drinking, stress, and depression on menstrual irregularities [14,15,16,17,18,19,20,21,22], as well as on menstrual disorders such as polymenorrhea, oligomenorrhea, and menorrhagia [23,24,25,26,27,28,29].

However, previous studies have predominantly focused on the association between lifestyle and psychosocial factors and the general menstrual cycle irregularities, with less attention on exploring the relationships of these factors with specific conditions, such as polymenorrhea, oligomenorrhea, and menorrhagia. Therefore, further research into the causes of these specific menstrual disorders is necessary to establish a scientific foundation for improving menstrual health. Additionally, in South Korea, the reports of health concerns related to menstrual pain and changes in menstrual flow due to sanitary pad use have increased the need to ascertain the menstrual status of women. This study aimed to examine the menstrual characteristics and associated factors among South Korean women aged 15–45 years using a nationally representative sample. Specifically, this study sought to determine the general menstrual characteristics and prevalence of specific menstrual disorders among Korean women while investigating the association of these disorders with lifestyle and psychosocial factors. The findings are expected to provide scientific evidence for the establishment of policies aimed at improving women’s health.

## 2. Materials and Methods

### 2.1. Data Collection and Preparation

Participants were recruited online and voluntarily provided informed consent to participate in accordance with the Declaration of Helsinki. The target population was South Korean women aged 15–45 years. The study population was strategically sampled based on the 2018 resident registration, considering the demographic distribution across 17 municipalities and provinces, to ensure representativeness. The age range was set to reflect periods of post-menarcheal menstrual stabilization and premenopausal menstrual irregularities. The study protocol was reviewed and approved by the Institutional Review Board of Hallym University Sacred Heart Hospital on 11 June 2019 (approval No. HALLYM 2019-04-023-001).

Data were collected from September to November 2019 using a sophisticated web-based survey system. The survey included validated questionnaires and multiple-choice questions to ensure clarity and reliability. All collected data were anonymized and stored securely to ensure the confidentiality of participants’ personal information.

The exclusion criteria were strictly applied to exclude the influence of external drugs or occupational exposures that affect menstruation, including oral contraceptives, users of intrauterine devices (also known as loops), pregnant or breastfeeding women, night-shift workers, individuals exposed to organic solvents or ethylene oxide gas at the workplace, and individuals with polycystic ovary syndrome. Night-shift work can disrupt circadian rhythm and hormone secretion [30,31], whereas exposure to organic solvents affects the endocrine system, causing menstrual disorders [32,33,34,35,36], necessitating their exclusion. Among 16,009 participants, 671 were excluded owing to exposure to night shifts, organic solvents, or ethylene oxide gas. Further exclusions were made for extreme outliers, including menstrual cycle length of <7 days (n = 104) or >365 days (n = 7), and those with a history of polycystic ovary syndrome (n = 1284), leaving 13,943 participants who were included in the final analysis (Figure 1). In addition, the cases with missing values were excluded and we used only the data available for each analysis.

### 2.2. Definition and Classification of Variables

Menstrual characteristics, such as menstrual cycle length (the number of days from the start of one period to the start of the next), menstrual duration (the number of days from the start to the end of menstruation), and menstrual disorders, and influencing factors, namely age, height, weight, education, income, smoking status, alcohol consumption status, depression, and stress, were surveyed. Furthermore, health and lifestyle variables such as BMI, smoking status, and alcohol consumption status were also assessed.

Among the menstruation-related variables are the menstrual cycle length, the menstrual duration, and menstrual disorders. Menstrual disorders were defined as follows: polymenorrhea, a menstrual cycle length shorter than 20 days; oligomenorrhea, a menstrual cycle length longer than 36 days; and menorrhagia, a bleeding period longer than 8 days.

Based on the height and weight indicated in the survey, BMI was calculated by dividing weight in kilograms (kg) by the square of height in meters (m^2^). Based on the BMI, the participants were categorized as underweight (<18.5 kg/m^2^), normal (18.5–24.9 kg/m^2^), overweight (25.0–29.9 kg/m^2^), and obese (≥30.0 kg/m^2^) according to the World Health Organization standards.

Regarding psychosocial factors, depression was assessed using the Patient Health Questionnaire-9 (PHQ-9) [37,38], and perceived stress was evaluated using the Perceived Stress Scale-10 (PSS-10) [39,40]. The PHQ-9 consists of nine items, each scored on a 4-point scale. The PHQ-9 score can range from 0 to 27 since each of the 9 items can be scored from 0 (not at all) to 3 (nearly every day). A total score of ≤4 is considered minimal, 5–9 is mild, 10–19 is moderate, and ≥20 is interpreted as severe depression. Participants with PHQ-9 scores of ≥10 were classified as a high-risk group for depression. The PSS-10 consists of 10 items assessing an individual’s psychological state over the past month. The PSS-10 score can range from 0 to 40 since each of the 10 items can be scored from 0 (never) to 4 (very often). A score of ≥13 is interpreted as low stress, 14–26 as moderate stress, and >26 as high stress. Participants with PSS-10 scores of ≥14 were classified as a high-risk group for stress. The Cronbach’s alpha, which represents internal consistency reliability, was 0.874 for PHQ-9 and 0.825 for PSS-10.

### 2.3. Statistical Analysis

The distribution of participants was described by general, psychosocial, and menstrual characteristics. The Chi-square test was employed to examine differences in the prevalence of menstrual disorders according to the participants’ general characteristics and psychosocial factors. In addition, using a linear combination of independent variables, a logistic regression analysis was performed to estimate the probability of an event, such as the occurrence of menstrual disorders. Adjusted odds ratios (ORs) of the risk factors for menstrual disorders were calculated after adjusting for other risk factors, including age, BMI, education, income, smoking status, alcohol consumption status, depression, and stress. Statistical analyses were performed using the SPSS software (version 27.0; IBM Corp., Armonk, NY, USA).

## 3. Results

The majority of participants were in their 30s (35.4%) and 20s (31.2%), with the 15–19 years age group being the least represented at 13.7%, as is apparent from Table 1. Regarding BMI, 70.1% of participants were within the normal range, 15.6% were underweight, and 14.3% were overweight or obese. Current smokers comprised 7.6% of the participants, and 36.5% were current alcohol consumers. The proportion of the participants in the high-risk depression and stress groups was 22.3% and 82.6%, respectively. Approximately 68.2% of the participants were either currently enrolled in a university or had graduated from a university. The most common household income range was 2–4 million Korean won, as reported by 34.8% of the participants, and 13.4% did not specify their household income.

Menarche occurred at age 12 or 13 in 48.5% and 14–17 years in 39% of the participants, as is apparent from Table 2. The mean age of menarche was 12.9 years. Regarding the menstrual cycle length, 67.4% of the participants reported cycles ranging between 26 and 31 days, with an average cycle length of 29.5 days. The majority of the participants (79.3%) had menstrual durations of 5 to 7 days. The prevalence of polymenorrhea, oligomenorrhea, and menorrhagia in menstrual disorders was 3.1%, 9.0%, and 5.4%, respectively.

### 3.1. Prevalence of Menstrual Disorders by Risk Factors

The prevalence of menstrual disorders—polymenorrhea, oligomenorrhea, and menorrhagia—was significantly high at younger ages, as seen in Table 3. The higher prevalence of menstrual disorders among individuals aged 15–19 is estimated to be due to unstable menstrual patterns common during adolescence. In addition, obesity was associated with a higher prevalence of polymenorrhea (4.3%) and oligomenorrhea (17.5%) than that observed with normal weight. Current smokers had a higher prevalence of polymenorrhea (5.0%) than observed with non-smokers. The high-risk group for stress had a higher prevalence of oligomenorrhea (9.4%) and menorrhagia (5.6%) than the normal group.

### 3.2. Risk Factors for Menstrual Disorders

The univariate analysis showed that increases in age, education level, and household income were associated with a significant decrease in the OR, as is apparent from Table 4. Being underweight (OR: 1.430, 95% CI: 1.115–1.833) or overweight (OR: 1.366, 95% CI: 1.024–-1.824) significantly increased the OR compared with being normal weight. Current smokers exhibited a significantly higher OR of 1.688 (95% CI: 1.257–2.267) than that exhibited by non-smokers. Regarding psychosocial factors, the high-risk group for depression had a significantly higher OR of 1.481 (95% CI: 1.201–1.827) than that exhibited by the normal group. However, current drinkers exhibited a significantly lower OR of 0.483 (95% CI: 0.383–0.609) than that exhibited by non-drinkers. In the multivariate analysis, the effect of age disappeared across all age groups, and statistically significant associations with BMI (except for underweight) and depression were not maintained. Smoking status was significantly associated with polymenorrhea, with current smokers having a higher OR of 1.516 (95% CI: 1.099–2.091) than that by non-smokers, whereas current drinkers had a significantly lower OR of 0.569 (95% CI: 0.442–0.733) than that by non-drinkers.

Univariate analysis showed that participants in their 30s and 40s compared with those with other ages exhibited a decreased OR, and a higher educational level was associated with a significant decrease in the OR, as is apparent from Table 5. Regarding BMI, significantly higher ORs were observed with being overweight (OR: 1.348, 95% CI: 1.134–1.603) or obese (OR: 2.279, 95% CI: 1.764–2.944) than having normal weight. Additionally, current smokers exhibited a significantly higher OR of 1.232 (95% CI: 1.004–1.513) than that exhibited by non-smokers. Multivariate analysis showed that the OR was significantly lower for participants aged 20–45 years than for those aged 15–19 years, and statistically significant associations with being overweight or obese were maintained. Regarding alcohol consumption status, current drinkers exhibited a significantly higher OR of 1.17 (95% CI: 1.020–1.343) than that exhibited by non-drinkers. Additionally, the high-risk groups for depression and stress exhibited significantly elevated ORs of 1.416 (95% CI: 1.236–1.621) and 1.248 (95% CI: 1.047–1.487), respectively, compared with those exhibited by the normal groups.

In the univariate analysis, increases in age, education, and income were associated with a significant decrease in ORs, as is apparent from Table 6. Current smokers exhibited a significantly higher OR of 1.341 (95% CI: 1.044–1.723) than that exhibited by non-smokers, and former and current drinkers exhibited significantly decreased ORs of 0.741 (95% CI: 0.588–0.932) and 0.576 (95% CI: 0.486–0.683), respectively, than that exhibited by non-drinkers. Regarding psychosocial factors, depression and stress significantly increased ORs in the high-risk group. In the multivariate analysis, the ORs were still significantly lower for individuals with a higher age and income than for those with a lower age and income. In contrast, the risk of menorrhagia was significantly higher in the high-risk group for depression than in the normal group.

## 4. Discussion

This cross-sectional study examined the menstruation status of South Korean women and analyzed the impact of lifestyle and psychosocial factors on menstrual disorders. First, the age at menarche was examined as a menstruation-related variable. Approximately half (48.5%) of the participants began menstruating at the age of 12 or 13 years, with an average age of menarche of 12.9 years. This finding is similar to the average age of menarche of 12.7 years among participants of the same age range in the 2021 Korea National Health and Nutrition Examination Survey [41]. We also examined the menstrual cycle length, and 67.4% of the respondents reported cycles ranging between 26 and 31 days, with 79% reporting menstrual durations of 5 to 7 days. Lawson et al. [42] conducted a cross-sectional analysis of 6309 nurses aged 21–45 years between 2010 and 2012 and found that 69.7% of participants had menstrual cycle lengths ranging between 26 and 31 days, aligning with the findings of this study. Our analysis showed that polymenorrhea, oligomenorrhea, and menorrhagia accounted for 3.1%, 9.0%, and 5.4% of the cases of menstrual disorders, respectively. Kulshrestha and Durrani [43] analyzed the prevalence of polymenorrhea (<21 days), oligomenorrhea (>35 days), and menorrhagia (>8 days) among adolescents aged 14–17 years and reported 22.2%, 12.8%, and 15.9% prevalences, respectively. Despite the lower values obtained in our study, this discrepancy may be due to differences in the age groups of the participants in this study and in the study by Kulshrestha and Durrani [43]. Notably, irregular menstruations are prevalent among adolescent girls [44], and irregular menstrual cycles are more common in younger teens (aged 12–14 years) than in older teens (aged 15–18 years) [45]. Generally, irregular menstruation among adolescents can be attributed to irregular ovulation intervals after menarche [46]. Moreover, it can be caused by various factors, such as excessive exercise, extreme changes in body weight, undesirable nutrition intake [47], and hormonal imbalances. In the aforementioned study by Lawson et al. [42], the prevalence of polymenorrhea (<21 days) was 1.5% and that of oligomenorrhea (32–50 days) was 13.2%. In comparison with the findings of our study, the prevalence of polymenorrhea was approximately twice as high and that of oligomenorrhea was approximately 1.5 times lower. This discrepancy may be attributed to the different definitions of oligomenorrhea; while it was defined as a menstrual cycle length of 36 days or longer in our study, Lawson et al. defined it as cycles ranging between 32 and 50 days.

Next, we examined the relationships between demographic, lifestyle, and psychosocial factors and menstrual disorders. The prevalences of oligomenorrhea and menorrhagia significantly decreased with increasing age, showing the same trend before and after adjustment for confounders. The prevalence of polymenorrhea significantly decreased with age in only the univariate analysis. Hahn et al. [23] also reported a decreasing trend in the prevalence of oligomenorrhea and menorrhagia with increasing age, albeit without statistical significance.

Regarding BMI, an increased prevalence of polymenorrhea was observed in the underweight group, and an increased prevalence of oligomenorrhea was observed in the overweight and obese groups. Similar results were obtained in the study by Hahn et al. [23], in which an increased prevalence of oligomenorrhea was observed in the obese group (BMI ≥ 30 kg/m^2^) in a cross-sectional study of 2613 nulliparous Danish women aged 18–40 years between 2007 and 2011. Similarly, in a cross-sectional study of 3941 premenopausal women aged 21–40 years in Iowa or North Carolina, Rowland et al. [24] found that the prevalence of oligomenorrhea was higher in the obese group (BMI ≥ 35 kg/m^2^) than in the other groups. Song et al. [25] conducted a cross-sectional study involving 9335 premenopausal nurses aged 22–45 years reported an increased prevalence of polymenorrhea among underweight nurses and an increased prevalence of oligomenorrhea among overweight (23 ≤ BMI < 25 kg/m^2^) or obese (BMI ≥ 25 kg/m^2^) nurses. Wei et al. [48] examined the relationship between BMI and menstrual cycle length among 726 Australian women aged 26–36 years and found a higher prevalence of oligomenorrhea among obese (BMI ≥ 30 kg/m^2^) individuals than among the other BMI groups, but without statistical significance. They also noted that an increasing BMI was significantly associated with high testosterone and free androgen index levels and low sex hormone-binding globulin (SHBG) levels. Moreover, they observed that high testosterone and free androgen index levels and low SHBG levels were associated with an increased OR for oligomenorrhea. Furthermore, Klenov et al. [49] suggested that obesity can adversely affect the hypothalamic-pituitary-ovarian axis by lowering the SHBG levels and elevating peripheral aromatization of androgens.

Regarding smoking status, the prevalence of polymenorrhea and menorrhagia was higher in current smokers than in non-smokers. Similarly, Rowland et al. [24] reported a correlation between the prevalence of polymenorrhea and the number of cigarettes smoked per day. Furthermore, Hahn et al. [23] reported an increased prevalence of polymenorrhea in smokers compared with non-smokers. Smoking influences hormonal secretion, synthesis, metabolism, distribution, and stimulation, thereby potentially shortening the menstrual cycle length [23,24]. Windham et al. [50] examined the impact of smoking on the hormonal function of women of childbearing age by measuring the levels of the pituitary hormone follicle-stimulating hormone (FSH) in 403 women and found that urinary FSH levels were approximately 30–35% higher in smokers than in non-smokers, suggesting that the chemicals in tobacco could affect endocrine regulation and the secretion of pituitary hormones, and potentially causing menstrual disorders.

Regarding alcohol consumption status, the prevalence of polymenorrhea and menorrhagia was significantly lower among current drinkers than among non-drinkers. Song et al. [25] reported a higher prevalence of polymenorrhea and a lower prevalence of oligomenorrhea in drinkers than in non-drinkers. Notably, previous studies exploring the relationship between alcohol consumption and menstrual disorders have reported varying results. Cooper et al. [27] examined the relationship between menstrual cycle length and alcohol consumption in 766 women aged 29–31 years and found that the OR for oligomenorrhea was significantly lower in current drinkers than in non-drinkers. Liu et al. [26] reported a reduction in the prevalence of oligomenorrhea among drinkers. Conversely, Schliep et al. [28], in their investigation of the influence of alcohol consumption on menstrual cycle length among 259 women aged 18–44 years, found no statistically significant correlations. The inconsistent results among the studies may be attributed to differences in alcohol type (wine, beer, etc.), consumption levels, drinking frequency, or variations in the sociodemographic characteristics and lifestyle of the participants.

Regarding depression, the high-risk group exhibited a higher prevalence of polymenorrhea and menorrhagia than exhibited by the normal group. In the aforementioned study by Rowland et al. [24], the prevalence of oligomenorrhea increased among those taking medications for depression, although the difference was not statistically significant. The prevalence of oligomenorrhea was higher in the high-risk group for stress. Chang et al. [29] explored the relationship between stress and menstrual cycles in a study of 1095 nursing students aged 18–25 years in Taiwan, categorizing mental stress into none (0 points), low (1–3 points), and high (≥4 points). They found that both low and high stress levels were associated with an increased prevalence of oligomenorrhea, and a high stress score was significantly associated with an increased prevalence of menorrhagia. A study involving 980 medical students by Alshayeb and Sakka [51] also reported a statistically significant relationship between perceived stress and oligomenorrhea. However, Chang et al. [29] and Alshayeb and Sakka [51] recruited nursing and medical students, respectively, who are more likely to be exposed to stressors such as academic and exam-related pressures, limiting the generalizability of their findings. Therefore, the present study is meaningful for examining the relationship between menstrual disorders and stress experienced generally by women aged 15–45 years.

On the other hand, previous studies have shown that depression and stress interfere with the reproductive hormones of the hypothalamic–pituitary–gonadal (HPG) axis, which is involved in reproductive competence [52,53]. When stressed, the hypothalamic–pituitary–adrenal (HPA) axis is activated, and corticotropin-releasing hormone (CRH) is secreted from the hypothalamic. It interacts with pituitary receptors to secrete adrenocorticotropic hormone (ACTH), which is transported through the blood and acts on the adrenal to release cortisol [54]. The cortisol secreted in this way interferes with the production of gonadotrophin-releasing hormone (GnRH), disrupting the release of luteinizing hormone and FHS and leading to irregular menstrual cycles [55,56].

This study had several limitations that must be acknowledged. First, this study employed a cross-sectional design, which does not allow for the determination of temporal relationships between lifestyle factors, psychosocial factors, and menstrual disorders. Further research is needed on the mechanisms by which sociopsychological factors influence menstrual disorders. However, this study is significant, as it provides national data on menstrual disorders. Second, owing to the nature of the self-administered questionnaire on menstrual characteristics, such as menstrual cycle length and duration, a potential response bias may exist. Third, even though the menstrual cycle is influenced by reproductive hormones, such as the luteinizing hormone, FSH, estrogen, and progesterone, this study did not verify whether the secretion of reproductive hormones in the participants was normal. The limitation in understanding the biological mechanisms underlying menstrual characteristics may potentially affect the validity of this study’s findings. Finally, since research data were available after the entire health effect survey, we analyzed data collected in 2019, which may limit the applicability of our findings to the current real-world situation.

Despite these limitations, this study is significant because it is a population-based study designed to ensure national representativeness, providing foundational data for understanding the menstrual characteristics of South Korean women aged 15–45 years. The results of our study can serve as useful data for improving menstrual disorders and raising awareness of menstrual disorders may help better manage women’s reproductive health.

## 5. Conclusions

Age, BMI, smoking status, depression, and perceived stress were identified as factors associated with menstrual disorders. This suggests that menstrual disorders may be reduced by improving BMI and smoking status, which are modifiable through individual efforts, such as dietary habit improvement, physical exercise, and smoking cessation. In addition to individual efforts, public health policies can improve the environmental and social factors influencing health behaviors. For instance, implementing community-based public health programs, such as a smoking cessation project at a community health center, a campaign to encourage exercise, and expanded subsidies for healthy food in the vulnerable population, could help in making these resources easily accessible and affordable for all women, including those from low-income backgrounds. Furthermore, a comprehensive approach that includes improving access to mental health services and medical care is needed. Given that depression and perceived stress generally require professional interventions, unlike lifestyle factors, managing psychosocial problems through counseling and medical consultations is imperative, underscoring the need to establish a social support system. Further research using a longitudinal design is needed to establish the causal relationships between various factors such as lifestyle and psychosocial factors and menstrual disorders. The results of this study are expected to be used as foundational data to enhance awareness of menstrual disorders and promote women’s health.

## Figures and Tables

**Figure 1 healthcare-13-00606-f001:**
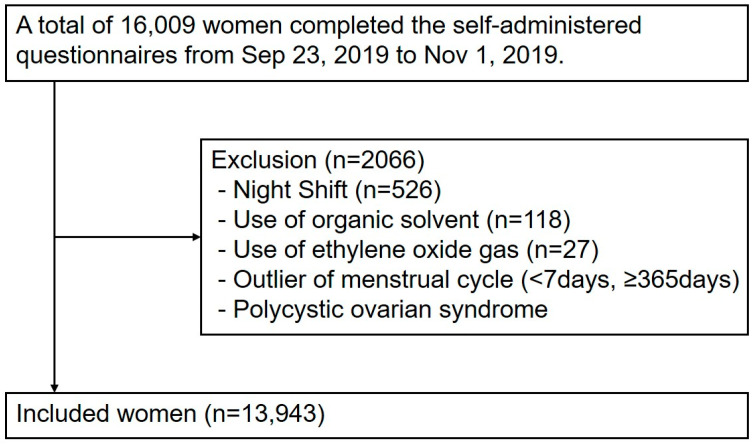
Flowchart for the eligibility of study participants.

**Table 1 healthcare-13-00606-t001:** General characteristics of participants (N = 13,943).

Variables	Number	Percentage (%)
Age (years)		
15–19	1917	13.7
20–29	4351	31.2
30–39	4937	35.4
40–45	2738	19.6
BMI (kg/m^2^)		
<18.5 (Underweight)	2172	15.6
18.5–24.9 (Normal weight)	9747	70.1
25.0–29.9 (Overweight)	1548	11.1
≥30.0 (Obesity)	439	3.2
Smoking Status		
Non-smoker	12,211	87.6
Former smoker	674	4.8
Current smoker	1058	7.6
Alcohol Consumption Status		
Non-drinker	6992	50.1
Former drinker	1861	13.3
Current drinker	5090	36.5
Education		
Under high school graduation	3648	26.2
Attending university or graduation	9504	68.2
Over graduate school	791	5.7
Income (Korean Won)		
<2,000,000	2490	17.9
2,000,000–4,000,000	4857	34.8
≥4,000,000	4722	33.9
Unsure	1874	13.4
Depression		
Minimal (0–4)	6286	45.1
Mild (5–9)	4543	32.6
Moderate (10–19)	2707	19.4
Severe (20–27)	407	2.9
Stress		
Low (0–13)	2422	17.4
Moderate (14–26)	10,336	74.1
High (27–40)	1185	8.5

Note. Missing value: BMI (n = 37). Depression was measured using the PHQ-9, and stress was measured using the PSS-10. BMI, body mass index; PHQ-9, Patient Health Questionnaire-9; PSS-10, Perceived Stress Scale-10.

**Table 2 healthcare-13-00606-t002:** Menstrual characteristics.

Variables	Number	Percentage (%)
Menarche age (years)		
≤8	11	0.1
9–11	2753	19.8
12–13	6882	49.5
14–17	4104	29.5
≥18	140	1.0
Mean ± S.D.	12.9 ± 2.2	
Menstrual cycle length (days)		
<21	436	3.1
21–25	1179	8.5
26–31	9394	67.4
32–35	1682	12.0
≥36	1252	9.0
Mean ± S.D.	29.5 ± 4.9	
Menstrual duration (days)		
<5	2128	15.3
5–7	11,063	79.3
≥8	749	5.4

Note. Missing values: menarche age (n = 53), menstruation period (n = 3), and menorrhagia (menstrual duration ≥ 8 days; n = 1). S.D., standard deviation.

**Table 3 healthcare-13-00606-t003:** The prevalence of menstrual disorders by general characteristics and psychosocial factors.

Unit: N (%)			
**Variables**	Polymenorrhea	Oligomenorrhea	Menorrhagia
Age (years)			
15–19	108 (5.6)	259 (13.5)	195 (10.2)
20–29	137 (3.1)	536 (12.3)	215 (4.9)
30–39	114 (2.3)	342 (6.9)	221 (4.5)
40–45	77 (2.8)	115 (4.2)	118 (4.3)
*p*-value	<0.001	<0.001	<0.001
BMI (kg/m^2^)			
<18.5 (Underweight)	85 (3.9)	166 (7.6)	507 (6.0)
18.5–24.9 (Normal weight)	270 (2.8)	832 (8.5)	131 (5.2)
25.0–29.9 (Overweight)	58 (3.7)	173 (11.2)	79 (5.1)
≥30.0 (Obesity)	19 (4.3)	77 (17.5)	28 (6.4)
*p*-value	0.005	<0.001	0.320
Smoking Status			
Non-smoker	370 (3.0)	1080 (8.8)	640 (5.2)
Former smoker	13 (1.9)	59 (8.8)	36 (5.3)
Current smoker	53 (5.0)	113 (10.7)	73 (6.9)
*p*-value	<0.001	0.131	0.070
Alcohol Consumption Status			
Non-drinker	276 (3.9)	610 (8.7)	459 (6.6)
Former drinker	61 (3.3)	167 (9.0)	92 (4.9)
Current drinker	99 (1.9)	475 (9.3)	198 (3.9)
*p*-value	<0.001	0.514	<0.001
Education			
Under high school graduation	211 (5.8)	383 (10.5)	293 (8.0)
Attending university or graduation	209 (2.2)	820 (8.6)	422 (4.4)
Over graduate school	16 (2.0)	49 (6.2)	34 (4.3)
*p*-value	<0.001	<0.001	<0.001
Income (Korean Won)			
<2,000,000	143 (5.7)	241 (9.7)	188 (7.6)
2,000,000–4,000,000	116 (2.4)	410 (8.4)	230 (4.7)
≥4,000,000	79 (1.7)	366 (7.8)	194 (4.1)
Unsure	98 (5.2)	235 (12.5)	137 (7.3)
*p*-value	<0.001	<0.001	<0.001
Depression			
Normal group (0–9)	307 (2.8)	869 (8.0)	510 (4.7)
High-risk group (10–27)	129 (4.1)	383 (12.3)	239 (7.7)
*p*-value	<0.001	<0.001	<0.001
Stress			
Normal group (0–13)	76 (3.1)	169 (7.0)	109 (4.5)
High-risk group (14–40)	360 (3.1)	1083 (9.4)	640 (5.6)
*p*-value	0.973	<0.001	0.036

Note. Missing values: BMI (n = 37), menorrhagia (n = 1). Depression was measured using the PHQ-9, and stress was measured using the PSS-10. BMI, body mass index; PHQ-9, Patient Health Questionnaire-9; PSS-10, Perceived Stress Scale.

**Table 4 healthcare-13-00606-t004:** Odds ratios for polymenorrhea by general characteristics and psychosocial factors.

Variables	Crude		Adjusted	
	OR	95% CI	OR	95% CI
Age (years)				
15–19	1.000		1.000	
20–29	0.545	0.421–0.705	1.037	0.755–1.425
30–39	0.396	0.303–0.518	0.958	0.684–1.343
40–45	0.485	0.360–0.653	1.191	0.833–1.701
BMI (kg/m^2^)				
<18.5 (Underweight)	1.430	1.115–1.833	1.291	1.003–1.661
18.5–24.9 (Normal weight)	1.000		1.000	
25.0–29.9 (Overweight)	1.366	1.024–1.824	1.175	0.876–1.576
≥30.0 (Obesity)	1.588	0.987–2.554	1.075	0.661–1.749
Smoking Status				
Non-smoker	1.000		1.000	
Former smoker	0.629	0.360–1.100	0.670	0.379–1.185
Current smoker	1.688	1.257–2.267	1.516	1.099–2.091
Alcohol Consumption Status				
Non-drinker	1.000		1.000	
Former drinker	0.825	0.622–1.093	0.944	0.702–1.269
Current drinker	0.483	0.383–0.609	0.569	0.442–0.733
Education				
Under high school graduation	1.000		1.000	
Attending university or graduation	0.366	0.301–0.445	0.516	0.407–0.655
Over graduate school	0.336	0.201–0.562	0.555	0.322–0.954
Income (Korean Won)				
<2,000,000	1.000		1.000	
2,000,000–4,000,000	0.402	0.313–0.516	0.468	0.361–0.607
≥4,000,000	0.279	0.211–0.369	0.356	0.265–0.479
Unsure	0.906	0.695–1.180	0.761	0.571–1.015
Depression				
Normal group (0–9)	1.000		1.000	
High-risk group (10–27)	1.481	1.201–1.827	1.247	0.994–1.563
Stress				
Normal group (0–13)	1.000		1.000	
High-risk group (14–40)	0.996	0.774–1.280	0.916	0.702–1.196

Note. Missing values: BMI (n = 37). Adjusted ORs were obtained after adjusting for all other variables in this table; Depression was measured using the PHQ-9, and stress was measured using the PSS-10. OR, odds ratio; CI, confidence interval; BMI, body mass index; PHQ-9, Patient Health Questionnaire-9; PSS-10, Perceived Stress Scale.

**Table 5 healthcare-13-00606-t005:** Odds ratios for oligomenorrhea by general characteristics and psychosocial factors.

Variables	Crude		Adjusted	
	OR	95% CI	OR	95% CI
Age (years)				
15–19	1.000		1.000	
20–29	0.899	0.767–1.055	0.825	0.673–1.010
30–39	0.476	0.402–0.565	0.436	0.350–0.543
40–45	0.281	0.223–0.353	0.261	0.201–0.340
BMI (kg/m^2^)				
<18.5 (Underweight)	0.887	0.745–1.055	0.801	0.672–0.955
18.5–24.9 (Normal weight)	1.000		1.000	
25.0–29.9 (Overweight)	1.348	1.134–1.603	1.354	1.134–1.615
≥30.0 (Obesity)	2.279	1.764–2.944	2.164	1.662–2.818
Smoking Status				
Non-smoker	1.000		1.000	
Former smoker	0.989	0.752–1.301	0.952	0.717–1.264
Current smoker	1.232	1.004–1.513	1.008	0.810–1.254
Alcohol Consumption Status				
Non-drinker	1.000		1.000	
Former drinker	1.031	0.862–1.234	1.066	0.883–1.286
Current drinker	1.077	0.950–1.221	1.170	1.020–1.343
Education				
Under high school graduation	1.000		1.000	
Attending university or graduation	0.805	0.708–0.915	1.071	0.910–1.261
Over graduate school	0.563	0.414–0.766	0.940	0.674–1.310
Income (Korean Won)				
<2,000,000	1.000		1.000	
2,000,000–4,000,000	0.860	0.728–1.017	1.123	0.944–1.335
≥4,000,000	0.784	0.661–0.930	1.161	0.969–1.392
Unsure	1.338	1.106–1.619	1.190	0.973–1.456
Depression				
Normal group (0–13)	1.000		1.000	
High-risk group (14–27)	1.607	1.415–1.826	1.416	1.236–1.621
Stress				
Normal group (0–13)	1.000		1.000	
High-risk group (14–40)	1.383	1.169–1.637	1.248	1.047–1.487

Note. Missing values: BMI (n = 37). Adjusted ORs were obtained after adjusting for all other variables in this table; Depression was assessed using the PHQ-9, and stress was assessed using the PSS-10. OR, odds ratio; CI, confidence interval; BMI, body mass index; PHQ-9, Patient Health Questionnaire-9; PSS-10, Perceived Stress Scale.

**Table 6 healthcare-13-00606-t006:** Odds ratios for menorrhagia by general characteristics and psychosocial factors.

Variables	Crude		Adjusted	
	OR	95% CI	OR	95% CI
Age (years)				
15–19	1.000		1.000	
20–29	0.459	0.375–0.562	0.565	0.439–0.728
30–39	0.414	0.339–0.506	0.557	0.430–0.723
40–45	0.398	0.314–0.504	0.551	0.414–0.732
BMI (kg/m^2^)				
<18.5 (Underweight)	1.170	0.960–1.426	1.066	0.872–1.302
18.5–24.9 (Normal weight)	1.000		1.000	
25.0–29.9 (Overweight)	0.980	0.768–1.250	0.905	0.707–1.157
≥30.0 (Obesity)	1.241	0.838–1.839	0.994	0.666–1.483
Smoking Status				
Non-smoker	1.000		1.000	
Former smoker	1.020	0.722–1.441	1.172	0.822–1.672
Current smoker	1.341	1.044–1.723	1.380	1.056–1.804
Alcohol Consumption Status				
Non-drinker	1.000		1.000	
Former drinker	0.741	0.588–0.932	0.792	0.624–1.005
Current drinker	0.576	0.486–0.683	0.637	0.529–0.767
Education				
Under high school graduation	1.000		1.000	
Attending university or graduation	0.532	0.456–0.621	0.842	0.692–1.024
Over graduate school	0.514	0.357–0.740	0.874	0.588–1.300
Income (Korean Won)				
<2,000,000	1.000		1.000	
2,000,000–4,000,000	0.609	0.499–0.743	0.699	0.569–0.859
≥4,000,000	0.525	0.427–0.645	0.648	0.520–0.807
Unsure	0.966	0.768–1.214	0.718	0.562–0.918
Depression				
Normal group (0–9)	1.000		1.000	
High-risk group (10–27)	1.683	1.435–1.973	1.521	1.284–1.802
Stress				
Normal group (0–13)	1.000		1.000	
High-risk group (14–40)	1.248	1.014–1.537	1.124	0.905–1.397

Note. Missing values: BMI (n = 37), menorrhagia (n = 1). Adjusted ORs were obtained after adjusting for all other variables in this table. Depression was measured using the PHQ-9, and stress was measured using the PSS-10. OR, odds ratio; CI, confidence interval; BMI, body mass index; PHQ-9, Patient Health Questionnaire-9; PSS-10, Perceived Stress Scale.

## Data Availability

The data presented in this study are available upon request from the corresponding author due to privacy restrictions. Requests to access the datasets be directed to the corresponding author.

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
