# Peer review of "Prevalence and Risk Factors of Menstrual Disorders in Korean Women"

_healthcare, 2025, doi:10.3390/healthcare13060606_

Round 1
Reviewer 1 Report
Comments and Suggestions for Authors
This is a cross-sectional study investigating the prevalence and risk factors of menstrual disorders in South Korean women aged 15-45. The authors then use an online questionnaire to gather data from a sample size of 13,943 participants, looking at various biological, lifestyle, and psychosocial factors. There are some issues that should be clarified and explained.
Abstract
1. The abstract does not include sufficient detail on the specific statistical methods utilized for analysis, which may detract from the perceived rigor of the study. The methods described herein should be made clearer to include, briefly, information on the main statistical techniques used: for instance, "Logistic regression analysis was utilized to determine odds ratios."
2. The abstract fails to provide background information on the importance of menstrual health, which may leave readers hazy as to the study's relevance. One additional sentence pointing to the significance of menstrual disorders within women's health would be of the essence, such as, "Understanding menstrual disorders is crucial for women's health and quality of life."
3. The abstract does not address the implications of the findings for public health or future research; this considerably weakens its impact. End with a statement like: "These findings can inform public health initiatives aimed at improving awareness and management of menstrual health among women."
Main text
Introduction:
4. The introduction is too broad, covering a wide range of topics on women's health without focusing on menstrual disorders. While the importance of women's health is established, the transition to menstrual health could be more direct and concise.
5. Thus, let the introduction be tailored to be more specific about menstrual health and its importance, rather than general women's health, to make it relevant.
6. While statistical data is given concerning the prevalence of menstrual disorders, the introduction does not appropriately contextualize why these disorders are particularly concerning for South Korean women. Therefore, include specific cultural or societal factors in South Korea that may exacerbate the impact of menstrual disorders, such as stigma or lack of healthcare access.
7. The introduction cites previous studies but fails to critically analyze their findings. There is no clear indication of the gaps in the literature or how this study intends to fill them. Therefore, give a more detailed review of relevant studies, outlining their limitations, and explicitly state how this research will fill those gaps in understanding menstrual disorders.
8. Certain statements, like those on women's health being linked to economic growth, are not only valid but too general and not specifically related to the focus of the study on menstrual disorders. Therefore, replace general statements with more specific ones detailing how menstrual health directly contributes to broader health outcomes and economic factors.
9. The research aim has been identified, but not clearly explained in terms of hypotheses or expected outcomes from the study. Therefore, the research questions and hypotheses should be clearly stated towards the end of the introduction to provide a background for the study's objectives.
Material and method:
10. Although the section provides information on the fact that data were collected through a web-based survey system, it lacks information on how the survey was designed, including question types and validation processes. Therefore, include a short description of the survey design, for example, "The survey included validated questionnaires and multiple-choice questions to ensure clarity and reliability."
11. The exclusion criteria are listed but not sufficiently justified; this may raise several questions about the comprehensiveness of the study. Also, provide a rationale for each exclusion criterion. Explain, for instance, the reasons why night-shift workers or those exposed to some kinds of chemical were excluded. Emphasize how these elements could confound the results.
12. The definitions of menstrual disorders and psychosocial factors are present but could be clearer or more comprehensive. Hence, the expansion of the definitions perhaps by including references to existing literature or guidelines that support the chosen classification.
13. In the statistical analysis section, it mentions t-tests and ANOVA, yet it never states why these tools were utilized or how it applies to the data being analyzed. Hence, detail the rationale for using particular statistical tests; for instance, "T-tests have been used in comparing means of two groups, while ANOVA was applied for comparisons of more than two groups regarding differences in menstrual disorder prevalence."
14. While the ethical approval issue is touched upon, the paper should also provide a much better account of how confidentiality of participants and data security was maintained. Therefore, one could have added a statement such as: "All data were anonymized and stored securely to protect participant confidentiality."
Results:
15. The presentation of the results, especially in the tables, is not clear enough. For example, the tables could be better formatted to make them more readable, thus enabling readers to get an overview of the data much faster. Therefore, clearer headings, consistent formatting, and perhaps color-coding or shading to differentiate between various categories within the tables should be used.
16. The section notes missing values for a number of variables but does not consistently report the number of missing values for all relevant variables across the tables. This inconsistency may cause confusion about the completeness of the data. Therefore, clearly indicate the number of missing values for each variable in the tables themselves, ensuring consistency across all data presentations.
17. Although the statistical significance is reported as represented by p-values, clinical significance regarding these findings is not adequately discussed. The reader cannot tell practically what these results imply. Therefore, briefly discuss the clinical relevance of the findings, especially those referring to prevalence rates and odds ratios.
18. The results are presented without sufficient context. For instance, discussing the implications of the psychosocial factors on menstrual disorders would add value to the data. Therefore, integrate a brief interpretation of the results within the section, explaining how these findings relate to existing literature and their potential impact on public health.
19. The problem of disconnection between narrative and tables occurs, since often there are some findings mentioned in the text without appropriately leading one through to the relevant table for detailed comprehension. Hence, explicit references to tables, such as "The great majority of the sample respondents were in their 30s, as is apparent from Table 1, should be used, giving a better flow in a discussion context.
20. It seems like there is an unnecessary repetition of some information. For instance, the prevalence rates of menstrual disorders are provided both in the text and the corresponding table without interpretation or analysis. Therefore, streamline the reporting by summarizing key findings in the text and referring to tables for detailed data, thus avoiding redundancy.
Discussion:
21. The discussion presents the findings; however, there is a lack of deeper analysis about the implications of these results. For example, although the associations with lifestyle and psychosocial factors are pointed out, the interaction and influence of these factors among themselves are not discussed. Hence, include a more detailed analysis regarding how lifestyle and psychosocial factors might interact with each other and their cumulative effect on menstrual disorders.
22. The section lacks critical analysis of the findings in light of the existing literature. Although comparisons are made, there is little discussion on why discrepancies exist or how they might inform future research. Therefore, provide a critical analysis of the results in relation to previous studies, discussing possible reasons for differences and what this means for understanding menstrual health.
23. While the conclusion mentions modifiable factors, it does not go further to give recommendations to practitioners or policymakers on how to address these issues. Therefore, give actionable recommendations based on the findings, such as specific interventions or public health campaigns that could be implemented to improve menstrual health.
24. The conclusion would then be that an improved BMI and smoking status could reduce menstrual disorders, without sufficiently deriving from such a conclusion the complexity of the issues involved or the need for comprehensive approaches. Therefore, Recognize the complexity of menstrual disorders and stress that lifestyle changes, while important, must be part of a more general approach, including mental health and healthcare access.
25. While limitations are mentioned, they are not emphasized and discussed in relation to the findings and conclusions that might be drawn from the study. Therefore, expand on limitations by discussing how they may impact the validity of results that can be generalized to other populations.
26. As stated, the discussion does not provide clear directions for further research, which is quite instrumental in developing the knowledge within this area. Therefore, include specific areas for future research, like longitudinal studies to establish the causal relationships or studies that may put much focus on different demographic groups.
Comments on the Quality of English LanguageThere are places where the language could be more specific and succinct. Consider revising sentences for clarity and accuracy of technical terms. Finally, a thorough proofreading or professional editing could help to enhance the overall quality of the manuscript.
Reviewer 2 Report
Comments and Suggestions for Authors
Dear Authors,
I sincerely thank you for the opportunity to review this manuscript, that I read with great interest. In my opinion it is quite well written and structured, I have some suggestions in order to further improve its quality.
- Line 16: I suggest to replace the word "what" with "which";
- Line 20: I suggest to replace "compared by" with "compared with";
- Line 28: I would specify which type of study is this manuscript;
- Lines 40-42: I suggest to add a citation or, if this sentence is an Authors' opinion, to move it to the Discussion section;
- Line 47: I suggest to better specify what do "short" and "long cycle" mean, in terms of days/weeks/months;
- Lines 49-51: are there more updated data about the prevalence of menstrual disorders in Sud Korea?
- Lines 84-85: why did the Authors chose to publish in 2025 a work based on data collected in 2019? This could be an issue, since the presented data might be obsolete;
- Lines 85-89: please add to the exclusion criteria the history of polycystic ovary syndrome, since it is mentioned in the following lines;
- Line 97: please specify what do you mean by mentioning the "menstrual cycle" among the menstrual characteristics, since currently it is not clear at all. Maybe the Authors could place in these lines the definition in line 110;
- Line 129: which scale did the Authors use to define a weight "normal" or not? Please quote it;
- Line 157: there is a little typo within the word "menstrual";
- In the overall Discussion section, I suggest to quote some systematic reviews, more scientifically valuable than other studies currently cited;
- Line 220: I suggest to replace the word "somewhat" with another word, more precise and suitable for a scientific work;
- Line 289: please mention the not updated data used for this research as a limitation of this study;
In conclusion, I personally believe that this manuscript can be reconsidered after major revisions. I remain at your disposal.
Best Regards
Reviewer 3 Report
Comments and Suggestions for Authors
Please amend the paper following the comments:
1. Title should be capitalised.
2. The introduction is concise and it is nice, however, previous studies should be mentioned in more detail, especially studies conducted in South Korea.
3. Adding more keywords would be beneficial for better indexing.
4. Please add the date of ethical approval.
5. Lines 75-77: check with the editor, whether this information should be presented here in the Data Availability section.
6. The recruitment process is described insufficiently. Please add the platform you used. How was the link distributed?
7. Please describe what do you mean providing data on "organic solvents, or ethylene oxide gas." as exclusion criteria. It is unclear.
8. How did you check the quality of the data? Please check at least whether minimal and maximum potential scores of the most critical variables (e.g., BMO) are appropriate. Did you use attention check questions?
9. Internal consistency reliability of the PHQ-9 and PSS-10 should be calculated. Questionnaire should be described, and references for Korean versions should be provided. Please use depression symptoms instead of just depression as PHQ-9 measures only depression symptoms. This will help to avoid overstatements.
10. Please avoid one-sentence paragraphs.
11. The study lacks practical implications for healthcare, however, the authors wanted "to enhance awareness of menstrual disorders and promote women’s health." based on the study's results. Practical implications should be clearly stated. As there are a lot of numbers in the discussion with a lot of data, it will be better to present the key results with corresponding recommendations in a form of a table.
12. Lines 312-313 were repeated in the abstract. I would suggest the authors to reconsider the same sentences in the different parts of the paper.
Reviewer 4 Report
Comments and Suggestions for Authors
Thank you for this interesting paper. Please find some suggestions/queries.
1. Introduction can benefit from studies/data related to menstrual disorders among adolescents as significant findings from the study also relate to the younger age groups.
2. Methods: A significant proportion of participants have less than graduate education. What measures were taken to make sure the participants understood technical language such as oligomenorrhea, PCOD, menorrhagia etc. Also, this data was obtained online, so this raises a concern on the comprehension of the questions posed to the participants.
3. Also, at younger ages, menstrual irregularities are common such as anovulatory cycles. How are these distinguished from disorders among the participants?
4. Were other bleeding disorders such as leukemia, platelet disorders, clotting factor deficiencies, or (less common) von Willebrand disease ruled out?
5. Results: I believe that some of the results have to be interpreted with caution. For eg: Low education being significantly associated with a menstrual disorder. Low education implies, lack of knowledge or having poor education, or lower literacy. This is to be expected in those in the age group of 15-19 who are being compared with those of older ages. In the life course perspective, these young participants may go on to acheive higher education. It would be pertinent for the authors to include statements to this effect.
6. References: Some of the references related to mensturation are from the 1990s. Please modify to updated references.
7. Ethics: The age range of participants is from 15 years onward, however, there is no mention of parental consent for adolescent participants or waiver procedures related to the same given that the study is online. Please clarify.
Reviewer 5 Report
Comments and Suggestions for Authors
The idea of ​​the article is very interesting and original, congratulations. My recommendations are the following:
Abstract
it would be recommended to draw a conclusion regarding the results.
Introduction
It would be of great interest to discuss the issue of sanitary pads (line 66), since in other contexts it is not understood what it refers to.
As a recommendation, it would also be of great interest to bring public attention to these phenomena in Korea. For example, in countries with high levels of conservatism or sexism, women's health phenomena are neglected and there is a lot of literature that exposes this.
Methods
It would be advisable to explain why the night shift is an exclusion criterion.
It is noteworthy that the reason for excluding the use of organic solvent or ethylene oxide gas is striking, whether it is medication, drug abuse or what.
It would be important to detail the internal consistency (Cronbach's alpha, for example) of the questionnaires used, such as those on depression and anxiety (lines 106-109).
Results
It would be interesting to see if the tables are all necessary, since there are several with similar data, and they may be redundant.
It would be important to review the format of the Tables since there are differential issues with APA 7th edition.
Discussion
It is advisable to refine the development of the discussion, reorganize it by items of interest and not start several paragraphs with "In this study" (lines 251 and 261), but "Regarding..." following the order of items. Ideally, it should be the same order as the presentation of results.
Conclusion
Although tobacco and BMI improvements are individual efforts, public policies must take responsibility. For example, if there is poverty, fast food and junk food are the most used resources, as shown by a lot of research. It is important to point out that every psychosocial variable is the consequence of an environment, and the individual has less weight than the social as a rule.
References
More relevant studies could be included that have not been considered. The references are somewhat short.
In line with the above, there is only one date for the year 2024.
Round 2
Reviewer 1 Report
Comments and Suggestions for Authors
1. Elaborate on how the psychosocial factors (e.g., depression and stress) may interact to determine menstrual disorders.
2. Discourse on the potential biological processes of psychosocial factors' contribution to menstrual disorders.
3. After all these exercises, a final round of editing is suggested to delve further into the sentence structure, grammar, and clarity. Some awkward points in V2 were:
Original: "The prevalence of menstrual disorders (polymenorrhea, oligomenorrhea, and menorrhagia) was significantly high at younger ages, as is apparent from Table 3."
Revised: "The prevalence of menstrual disorders—polymenorrhea, oligomenorrhea, and menorrhagia—was significantly higher in younger age groups, as seen in Table 3."
Original: "This study had certain limitations."
Revised: "This study has several limitations that must be acknowledged."
4. Construct actionable steps for policymakers and practitioners.
5. Explain in one paragraph why particular statistical tests (e.g., chi-square test, logistic regression) were used, and explain how these tests applied to the analysis conducted.
6. A paragraph should be added addressing how missing values were handled and whether they may have influenced the analysis.
7. Add insight into South Korean culture and society toward factors that could have an impact on menstrual health.
8. Add discussion points about future directions for research in the conclusion section.
9. The terminology used must be consistent throughout the text. For example:
-Use "menstrual cycle length" when discussing days instead of "menstrual cycle."
-Keep using high-risk groups for depression, not depressed groups.
10. Complete a final round of proofreading just to check grammar, typos, and discrepancies. For example:
Original: "The prevalences of oligomenorrhea and menorrhagia significantly decrease..."
Revised: "The prevalences of oligomenorrhea and menorrhagia significantly decrease..."
Reviewer 2 Report
Comments and Suggestions for Authors
Dear Authors,
I sincerely thank you for the opportunity to review this manuscript in its revised version.
In my opinion this work has been significantly improved, although I still have certain doubts. First of all, the fact that tha data presented in this work were collected in 2019: the Authors wrote that "the study data became available only after the release of a 2023 press briefing", but that still does not explain why these data were submitted more than one year after 2023.
In addiction, I suggested to add some systematic reviews to the Discussion section and you answered me that "we couldn’t find systematic reviews related to this study’s topic and complement the overall Discussion". On the other hand, I performed a brief research and quickly found several systematic reviews thast could be cited in this work:
1) Anthon C, Steinmann M, Vidal A, Dhakal C. Menstrual Disorders in Adolescence: Diagnostic and Therapeutic Challenges. J Clin Med. 2024 Dec 16;13(24):7668. doi: 10.3390/jcm13247668. PMID: 39768589; PMCID: PMC11678717;
2) Waghmare BV, Jajoo S. Navigating the Challenges: A Comprehensive Review of Adolescent Gynecological Problems. Cureus. 2024 Mar 14;16(3) : e56200. doi: 10.7759/cureus.56200. PMID: 38618317; PMCID: PMC11016329;
3) Hobbs AK, Cheng HL, Tee EYF, Steinbeck KS. Menstrual Dysfunction in Adolescents with Chronic Illness: A Systematic Review. J Pediatr Adolesc Gynecol. 2023 Aug;36(4):338-348. doi: 10.1016/j.jpag.2023.05.005. Epub 2023 May 14. PMID: 37192680;
4) Lingsha Wu, Jing Zhang, Jie Tang, Haiyan Fang. The relation between body mass index and primary dysmenorrhea: A systematic review and meta-analysis. https://doi.org/10.1111/aogs.14449).
These results sincerely make me worry about the overall research methodology of this manuscript, thus I hope that the Authors can perform an overall and (this time) actually precise revision of all the citation they added to the bibliography, in order to add more scientifically valuable works and to include only the most proper ones.
Best Regards
Reviewer 3 Report
Comments and Suggestions for Authors
Please find the comments:
1. Section 2.1. Study Participants describes the procedure of the study, not only the participants. Moreover, the results also contains the participant descriptions. These sections are overlapping. Please restructure the paper.
2. Comment 9 was not addressed:
Comments 9: Internal consistency reliability of the PHQ-9 and PSS-10 should be calculated.
Questionnaire should be described, and references for Korean versions should be provided.
Please use depression symptoms instead of just depression as PHQ-9 measures only
depression symptoms. This will help to avoid overstatements.
Response 9: We reviewed the literature for the reliability and validity for PHQ-9 and PSS-10
in method section, and described high risk group.
I invite the authors to describe the measures according to basic standards in the field. Please indicate the full range of descriptions, including titles, number of questions with their examples, subscale if applicable, response scale, screening characteristics for the PHQ-9, possible range score, and interpretation of scores. Please indicate references for Korean versions of these measures and briefly mention their psychometric properties.
Also, internal consistency reliability coefficients (Cronbach's alpha and/or McDonald's omega) should be calculated for the PSS-10 and PHQ-9 in these data. This is a very basic requirement for studies included psychometric questionnaires.
3. Please justify the use of univariate analysis instead of multivariate analysis in the study. multivariate analysis is more comprehensive and should be used here to reveal relevant associations between factors. The sample size is high, therefore, it is better to use comprehensive methodology.
Minor:
Please check the citation format. For instance, here "This may be due to differences in the age groups of the participants in this study and of those in the study by Kulshrestha and Durrani. In the aforementioned study by Lawson et al., the prevalence of polymenorrhea (< 21 days) was 1.5% and that of oligomenorrhea (32-50 days) was 13.2%".
